# Genomics of Adaptations in Ungulates

**DOI:** 10.3390/ani11061617

**Published:** 2021-05-29

**Authors:** Vivien J. Chebii, Emmanuel A. Mpolya, Farai C. Muchadeyi, Jean-Baka Domelevo Entfellner

**Affiliations:** 1School of Life Science and Bioengineering, Nelson Mandela Africa Institution of Science and Technology, P.O. Box 447, Arusha, Tanzania; emmanuel.mpolya@nm-aist.ac.tz; 2Biosciences Eastern and Central Africa, International Livestock Research Institute (BecA-ILRI) Hub, P.O. Box 30709, Nairobi 00100, Kenya; j.domelevoentfellner@cgiar.org; 3Agricultural Research Council Biotechnology Platform (ARC-BTP), Private Bag X5, Onderstepoort 0110, South Africa; MuchadeyiF@arc.agric.za

**Keywords:** ungulates, genome evolution, adaptive evolution, selection signatures

## Abstract

**Simple Summary:**

Ungulates are essential sources of food, labor, clothing, and transportation for humans. Due to their diverse habitats, ungulates have unique adaptive traits for survival. The availability of genomics resources has made it possible to identify genes associated with adaptations of ungulates to their respective environments. In this review, we document available ungulate genomics resources and genes associated with adaptive traits.

**Abstract:**

Ungulates are a group of hoofed animals that have long interacted with humans as essential sources of food, labor, clothing, and transportation. These consist of domesticated, feral, and wild species raised in a wide range of habitats and biomes. Given the diverse and extreme environments inhabited by ungulates, unique adaptive traits are fundamental for fitness. The documentation of genes that underlie their genomic signatures of selection is crucial in this regard. The increasing availability of advanced sequencing technologies has seen the rapid growth of ungulate genomic resources, which offers an exceptional opportunity to understand their adaptive evolution. Here, we summarize the current knowledge on evolutionary genetic signatures underlying the adaptations of ungulates to different habitats.

## 1. Introduction

Ungulates are hoofed animals that are either odd-toed, such as the horse, donkey, and rhinoceros, or even-toed, such as pigs, goats, camels, cattle, deer, and giraffes (see Figure 1) [1]. Other even-toed non-hoofed ungulates include cetaceans such as dolphins, whales, and porpoises [2]. Domesticated ungulates such as cows, sheep, and goats are an essential source of food and livelihoods to humans worldwide. In contrast, the wild ungulates generate huge returns through eco-tourism and sport hunting. Ungulates inhabit diverse environments, including some of the most unfriendly environments on earth, such as camels, reindeer, and wild goats in hot deserts [3,4,5], giraffes in tropical savanna plains [6,7], and yaks and Tibetan antelopes in high-altitude grasslands [8,9]. Yakutian horse, reindeer, and the woolly mammoth are endemic to arctic/sub-arctic forests [10,11,12,13], while whales and dolphins inhabit marine environments [14,15,16]. The diverse selective pressures in ungulates habitats have resulted in genetic changes that favor particular adaptive phenotypes [5,16]. For example, adaptive evolution of energy-related genes has been reported in desert-adapted camels, a possible adaptation to starvation [5]. Besides the wild ungulates, domesticated ungulates, such as cattle, goats, and pigs, also show adaptations to domesticated environments, such as ranching and intensive zero-grazing systems [4].

Despite the economic importance of ungulates, their adaptive evolution in different habitats is yet to be completely understood. The detection of genes underpinning their genomic signatures of selection and evolution is crucial to comprehend their adaptations to different habitats [21]. This review assesses the adaptive evolution of ungulates with a closer look at their adaptive evolutionary mechanisms and signatures in different habitats using comparative genomics. The review further highlights the future perspectives and directions concerning ungulate genomics and adaptive evolution. Together, such information can be useful in informing future genomics research in ungulates and other mammalian species.

## 2. Understanding the Genetic Basis of Ungulate Adaptations

It is paramount to identify loci associated with adaptive phenotypes to understand the genetic basis of adaptation of a given species to its environment [22]. Genomic regions controlling adaptations (selection signatures) are routinely detected using population genomics and comparative genomics-based approaches. Population genomics-based approaches at the disposal of researchers include: population differentiation (FST) [23], frequency-based methods (Tajima’s D) [24], haplotype-based methods (EHH) [25], and composite methods [26]. Each of the approaches detects different genetic signals which contribute to positive or negative selection as outlined in Table 1.

Comparative genomic approaches identify sequences that code for proteins or are conserved among different species and then search for lineage-specific accelerations in the rate of evolution. Lineage-specific accelerations are defined as an excess of substitutions compared to the baseline mutation rate, and is calculated from the overall rate of substitutions between species. Comparative approaches include; Hudson–Kreitman–Aguadé (HKA) test [27], McDonald–Kreitman test (MKT) [28], and Ka/Ks statistics [29]. The Hudson–Kreitmann–Aquadé test (HKA) determines the ratios of fixed interspecific differences to within-species polymorphisms (P) across loci [27]. McDonald–Kreitman test (MKT), on the other hand, compares the amount of variation within a species to the divergence between species at two types of sites (synonymous and non-synonymous) [28]. The most widely used comparative approach for detecting selection signatures between species is the dN/dS ratio, also known as the Ka/Ks statistic [29]. The non-synonymous/synonymous ratio (ω = dN/dS), also known as Ka/Ks statistic, identifies function-altering mutations by estimating the dN and dS ratio [29]. Where ω < 1 indicates negative (purifying) selection, ω = 1 indicates neutral evolution, and ω > 1 indicates positive (adaptive evolution) [29]. The ω ratio summarizes evolutionary rates of genes and provides an informative overview of the least (most) conserved genes and genes that have undergone adaptive evolution.

Other selection signatures detected using comparative genomic analysis include structural variants, such as deletions, duplications, and insertions. Several structural variants such as copy number variants are now detectable from short and long read sequence data. Copy number variants (CNVs) are genomic structural variants involving duplications or deletions of segments greater than 1000 bp, leading to copy number differences among individuals within or between species [30]. CNVs confer phenotypic effects by changing gene dosage, transcript structure, or regulating genes’ expression and functions, linked to adaptations in species [30,31]. Recent advances in genome sequencing technologies have significantly boosted adaptive phenotypes’ detection using either population or comparative genomic-based approaches. In this regard, we highlight some of the successes made in detecting adaptive loci in ungulates using either population or comparative genomics.

Advances in next-generation sequencing have seen the release of several desert ungulate genomes, such as red deer [3,32,33] and camels [5,34,35,36,37,38,39]. The genome sequences of domesticated ungulates, such as cattle [40,41,42,43], domestic horse [44], domestic goats [45,46,47,48,49], water buffalos [50,51], domestic pigs [52], domestic sheep [53,54,55], donkeys [56], and camels [5], have been archived in public biological databases. Similarly, genome sequences of wild ungulates from the artic and sub-arctic habits, such as reindeers [12,57,58] and Yakutian horses [11], savanna grasslands, such as giraffes [6], Tule elks [59] and bisons [60], high-altitude environments, such as Tibetan wild boars [52], yaks [8,61], Forest musk deer [62], Tibetan antelopes [9], and Siberian musk deer [63], and hot humid equatorial forests, such as okapis [6], are readily available in the public domain. The genomes of aquatic ungulates, such as dolphins [15,64] and whales [14,65,66,67], have also been sequenced. A summary of the available ungulate genomes is provided in Table 2. Advancements in population and comparative genomics and the increased availability of whole genome sequences have led to the identification of adaptive signatures in ungulates.

## 3. The Adaptive Evolution of Ungulates in Different Habitats

Adaptive evolution occurs when a species reacts to challenges posed by changes to its environment. Ungulates are distributed across extensive habitats, including those with different latitudes, altitudes, and ecological climates. The selective pressures in their various habitats have driven genetic changes that favor specific phenotypes over generations for adaptation [68]. Genomics has primarily been used to explore the genetic basis of ungulate adaptation in different environments by detecting selection signatures. The availability of genomic resources has opened up new opportunities for comprehending the genetic basis of ungulate adaptations to various habitats. In Section 3.1 to Section 3.6, we discuss these adaptive signatures of evolution in ungulates across different habitats.

### 3.1. Arid and Semi-Arid Habitats

The world’s arid and semiarid regions are largely characterized by intense solar radiation, high temperatures, and limited water and food supply [34,69]. The long-term exposure of ungulates to strong ultraviolet (UV) radiation in desert environments can cause ophthalmic or skin damage [3,5,70]. It is hypothesized that desert ungulates have evolved to survive in their environment, and their genomes possibly contain signatures of these critical adaptations [69]. Identifying the genes and pathways that desert ungulates possess can enhance the understanding of the genetic mechanisms that enable them to survive in such inhospitable environments [36].

Desert ungulates, such as the Dromedary camel, Bactrian camel, and Tarim red deer, are exposed for extended periods to high UV radiation, possibly leading to ophthalmic conditions [3,5]. Several genes controlling photoreception and visual protection, such as *OPN1SW*, *CX3CR1*, and *CNTFR*, are under positive selection in desert-adapted camels [5]. Additionally, the positive selection of visual protection genes (*LAMB1*, *LAMB2*, *CYC*, *FANCF*, and *GPR98*) has been reported in the desert-adapted Tarim red deer [3]. Desert ungulates are prone to UV-induced oxidative stress, which leads to DNA damage. In consequence of the adverse effects of prolonged exposure to UV radiation, desert ungulates have evolved protective mechanisms against the ionization radiation as evidenced by positive selection of DNA repair genes (*SLX4*, *FANCF*, *FANCG*, *FANCI*, *ATR*, *ERP44*, *NFE2L2*, and *MGST2*) in Bactrian and Tarim red deer [3,5]. Several other DNA repair genes (*PMS1*, *SPO11*, *RAD54L*, *MUTYH*, *CHEK2*, *POLR2D*, and *CMPK1*) are also under positive selection in desert sheep [71]. Comparative genomic analysis of wild African goat (Nubian ibex) showed that skin barrier development and function genes, such as *ABCA12*, *ASCL4*, and *UVSSA*, were under positive selection, suggesting that the desert goats have evolved skin protection strategies against the damaging solar radiation [72].

Food scarcity is another major challenge faced by desert ungulates, and as such, the metabolism of energy/nutrients is crucial for their survival [71]. Comparative genome analysis of different camel species and closely related species have previously reported the expansion of *ACC2*, *DGKZ*, and *GDPD4* genes, which are linked to lipid/fat metabolism for energy production and storage in the Bactrian camel genome, probably as an adaptive response to food scarcity in their habitats [5]. Selection signatures spanning several *MYH* genes that have direct roles in energy metabolism were identified in comparative genome studies of desert goats and sheep and are thought to facilitate muscular contraction as an adaptation/response to the need for resilience while trekking long distances in search of food and water [69]. The selection of candidate genes such as *BIN1*, *MSTN/GDF8*, and hypoxia-induced factors (*HIF-1*) in the Egyptian fat-tail sheep was similarly linked to the need to adapt/respond to the oxygen debt and hypoxia-like conditions which may occur in the skeletal muscles during the long-distance travels in the desert to get food/water [71].

Desert ungulates, such as Bactrian camels, have an excellent water reservations mechanism as evidenced by the upregulation of aquaporin genes *AQP1*, *AQP2*, and *AQP3* in the renal cortex and medulla of its kidney, suggesting a unique strategy for efficient water reabsorption [5]. The positive selection of the oxytocin signaling pathway genes (*CALM2*, *COX2*, *KCNJ5*, and *CACNA2D1*) and arachidonic acid metabolism pathway genes (*GPX3*, *ANXA6*, *PTGS2*, and *GPX7*), which are both functionally associated with the regulation of water retention/reabsorption in the kidney, has been identified in the desert sheep (*Ovis aries*) as a possible adaptive response to water scarcity [55]. Unlike the other mammals, the Bactrian and dromedary camel showed an expansion of *NR3C2* and *IRS1* genes, which are critical in the reabsorption of sodium and the subsequent maintenance of water balance [5]. An expansion of *CYP2E* and *CYP2J* genes, which transforms arachidonic acid into 19(S)-HETE, has been reported in the Bactrian camel genome when compared to genomes of other mammals [36]. 19(S)-HETE is a vasodilator of renal preglomerular vessels that stimulate water reabsorption; hence, it is useful for survival in deserts [73]. The adaptation of ungulates to desert/arid environments is complex and involves numerous biological processes and traits that contribute to phenotypic variations seen in desert ungulates [69]. Some of the candidate genes selected for in ungulates adapted to desert habitats are summarized in Table 3.

### 3.2. High-Altitude Habitats

High-altitude habitats are extremely inhospitable environments and are characterized by low oxygen pressure, cold temperatures, and high intense solar radiation [9,74], which impose severe selective pressure on the ungulate species in these habitats [75]. Hypoxia resulting from the low barometric pressure in high altitudes is a key challenge to ungulates. Several distinctive hypoxia-related selection signatures have been reported in high-altitude ungulates, suggesting their unique adaptive mechanisms to hypoxic conditions. Adaptation to high-altitude conditions has been analyzed by conducting genome-wide scans for natural selection signatures in ungulates such as the Tibetan yak, sheep, cow, antelope, and several others. Genome sequence analysis of the Tibetan yak showed an expansion of hypoxic-related genes (*ARG2*, *MMP3*, and *ADAM17*) compared to the low-altitude cow [8]. Other positively selected hypoxia-related genes that have been identified in high-altitude ungulates include *ALB*, *ECE1*, *GNG2*, and *PIK3C2G* in the Tibetan wild boar [52], *GHR*, *BMP15*, *ZEB1*, and *CPLANE1* in the Tibetan sheep [76], and *PIK3R2*, *CUL5*, *EPAS1*, *RNF4*, *RNF7*, *TNFSF10*, *HIF1A*, *PDE1A*, *FRAP1*, *PDE3*, *TXN*, *PRMT1*, *NOX4*, *EIF4E*, and *SPSB2* in the Tibetan pig [77]. Additionally, four hypoxia-related genes (*ALB*, *ECE1*, *GNG2*, and *PIK3C2G*) with strong selective sweep signals have been reported in the Tibetan wild boar genome [52].

Genome sequence analysis of high-altitude ungulates, such as Tibetan sheep [78], Tibetan horse [79], Tibetan yak [80], Tibetan pig [77], and Tibetan cashmere goat [48], have shown that the hypoxia-related gene EPAS1 is under accelerating evolution, suggesting a possible adaptation to hypoxic conditions in the plateau. Other hypoxia-related genes under positive selection, such as *SPON1*, *DKK2*, and *JAZF1*, have been reported in Tibetan cattle, sheep, and goats, suggesting a convergent evolution at the molecular level in high-altitude ungulates [81]. Signals of adaptive evolution in the energy metabolism gene *PKLR* and the hypoxia-related gene *NOS3* have been reported in the Tibetan antelope in comparison to plain-dwelling mammals [9]. The analysis of selection signatures in cattle of high-altitude habitats has also revealed novel potential selection targets. Recent whole-genome sequence analysis of high-altitude Pakistani indicine cattle reported enrichment of genes belonging to the *HIF1* pathways, highlighting the importance of response to hypoxia at high altitudes [82].

Apart from hypoxic conditions, animals that live at high altitudes are also challenged by intensive UV radiation, which induces DNA damage [83]. The comparative analysis of the Tibetan wild boar and Duroc pig genomes established that the genome of the Tibetan wild boar has an expansion of gene families linked to DNA replication, binding, and integration (*ERCC4*, *BCL3*, *ERCC6*, *USF1*, *ZRANB3*, and *REV1*), probably as a response to maintain genomic stability and integrity during replication under extreme solar radiation [52]. Positive selection of DNA repair genes (*ATR*, *NEK4*, *EYA1*, *XRCC1*, *CNOT8*, *TRIP12*, *TOPBP1*, *ZFYVE26*, *PLA2R1*, *UIMC1*, *FBXO18*, and *MCM10*) have also been reported in Siberian musk deer relative to other musk deer that do not inhabit high-altitude environments [63].

Besides the hypoxic conditions and intense UV radiation, ungulates inhabiting high-altitude environments face food scarcity due to short growing seasons [84]. Ungulates inhabiting high altitudes exhibit enhanced nutritional capability and energy metabolism as an adaptive response to low food availability [85]. Genes such as *CAMK2B*, *GENT3*, *HSD17B12*, *WHSC1*, and *GLUL*, linked to various nutrition pathways, are under positive selection in the yak genome compared to other mammals, suggesting their importance in enhancing food acquisition at high altitude [8]. High-altitudes ungulates are faced by frequent fluctuations of extreme solar radiation and low-light conditions. Positive selection of retinoid-X-receptor binding genes (*MED24* and *NROB2*) in Siberian musk deer that inhabit Asian mountains and forests is thought to help them forage in darkness in their habitat [63]. An earlier analysis of the forest musk genome also established the presence of several PSGs, such as *GRK7*, *SAG*, *SLC24A1*, *RDH10*, *CYP26A1*, *RDH12*, *SDR16C5*, and *DGAT1*, enriched in the retinol metabolism and phototransduction pathways, probably as an adaptation to the low-light and night-time environments [62]. Overall, ungulates demonstrate different adaptive responses to high altitudes and some of the candidate genes under selection in ungulates adapted to high-altitude habitats are summarized in Table 3.

### 3.3. Savanna Habitats

The giraffe is one of the terrestrial ungulates with remarkable diversity in size, diet, and habitat [86]. Giraffes are prone to viral and bacterial infections in their savanna habitats; hence, they have evolved to cope with invading pathogens [87]. For instance, immune response genes such as Toll-like receptor genes are under positive selection in the giraffe, suggesting their evolutionary adaptation infectious pathogens in the savanna [86].

Giraffes have long legs and necks, which enable them to feed efficiently on tall acacia trees in savanna habitats. Adaptive evolution studies have shown that genes involved in skeletal system development, such as *HOXB13* and *FGFRL1*, are under positive selection in the giraffe, suggesting that it may have evolved elevated physique for feeding on the tall acacia trees [6]. Giraffes’ elevated stature calls for an excellent cardiovascular system to regulate blood pressure over a height of 6 m. Genome sequence analysis has shown that cardiovascular function genes such as adrenergic receptors α1 and β-2, urotensin-2b, and angiotensin-converting enzyme are under adaptive evolution in the giraffe, suggesting that they have evolved to sustain the relatively high blood pressure (2.5× relative to that of humans) [6].

Visual cognition is vital for the survival of terrestrial animals since it is required for acquiring food, avoiding predators, and recognizing mates, among other functions [88]. Giraffes have excellent aerial vision, partly attributed to their exceptionally long legs and neck [89]. However, recent comparative analysis has shown that the giraffe exhibits a positive selection of the *OPN1LW* gene that affects optical transparency and light-signaling pathways, which may also facilitate its vision [7].

### 3.4. Marine Habitats

Marine/aquatic habitats are largely inhabited by cetaceans, including whales, dolphins, and porpoises [15]. Cetaceans are closely related to terrestrial ungulates; however, they exhibit secondary adaptation to marine life after re-colonizing the water from land some million years ago [90]. Comparative genome analyses of cetaceans have provided important insights into their genomic determinants and traits of aquatic specializations [91] marked by adaptations to the physiological stresses from lack of oxygen, elevated quantities of reactive oxygen species (ROS), and high salt levels [67]. Fossil evidence suggests that cetaceans evolved from artiodactyls and possibly underwent major dietary changes during their transition from terrestrial to aquatic life [92], from herbivory to carnivory, whose major nutritional components are proteins and lipids [16]. Adaptive evolution of the cetacean gastric *PGA* gene that enables them to better digest proteins have been reported [92]. Similarly, signals of positive selection in digestive proteinases and lipases (*CPA1*, *PRSS1*, *CTRC*, and *CELA3B*) have been reported in cetaceans, unlike other mammalian species [92].

Swimming and diving in water require energy; thus, cetaceans use a lot of energy to support muscle contraction [16]. Positive selection of genes associated with motor activity and muscle contraction, such as *SCN4A*, has been reported in the dolphin genome relative to other mammals, such as the cow, dog, and human [16]. Efficient locomotion of cetaceans in water is also essential for foraging but is energy-consuming [93]. The positive selection of genes involved in aerobic respiration (*CS*, *MDH1B*, and *SDHA*) and anaerobic respiration (*HK1*, *PGAM2*, *PFKFB1*, and *OGDH*) has been reported in the dolphin genome [16]. Cetaceans have a thick fat layer, which provides them with thermal insulation and buoyancy during swimming [94]. Selection signals in lipid synthesis and metabolism genes (*APOA2*, *APOO*, *APOC4*, *FABP4*, *CCDC129*, *SERINC4*, *PLA2G5*, *RARRES2*, *PNLIPRP3*, and *NR1I3*) has recently been reported in the dolphin genome [92]. Genes involved in lipids/fatty acid biosynthesis (*DGAT1*, *ELOVOL2*, *ACSM3*, and *ELOVL5*) and those involved in lipid/fatty acid transport and localization (*APOA2*, *ANXA1*, and *ATP8B2*) have similarly been shown to be significantly enriched in the dolphin genome [16]. Likewise, the loss of epidermal genes such as *GSDMA*, *TGM5*, *DSG4*, and *DSC1* in dolphin genome may have contributed to hair loss, which minimizes drag while swimming [16]. 

Perhaps the most marked adaptation of marine ungulates is hypoxia response during deep-diving [95,96]. Under hypoxic conditions, reactive oxygen species (ROS) are generated by several cellular mechanisms; hence, an efficient detoxifying system is essential for marine ungulates’ survival. Glutathione metabolism pathway genes, such as *GPX2*, *ODC1*, *GSR*, *GGT6*, *GGT7*, *GCLC,* and *ANPEP*, are positively selected in whales and dolphins, suggesting that they have evolved an antioxidant system to combat the damaging effects of hypoxia-induced ROS [67]. Several other hypoxia-tolerance-related genes (*HBA*, *HBB*, *MB*, *EDN1*, *EDN2*, *EDN3*, *EDNRA*, *EDNRB*, *ADRA1D*, and *AVP*) have been reported to be under adaptive evolution in cetaceans, suggesting that they have evolved a mechanism to cope with hypoxic conditions [97]. In addition to positive gene selection patterns, the loss and/or inactivation of genes due to relaxed selection of a function that subsequently became obsolete has also been implicated in cetacean traits and unique adaptations to aquatic life [98]. For example, the loss of the *AMPD3* gene, which is expressed in the erythrocytes of the sperm whale, is associated with enhanced oxygen transport during diving [99]. Other genes that may have been lost in cetaceans during their transition from land to water include those that reduce the risk of thrombus formation during diving (*F12* and *KLKB2*) and lung inflammation from oxidative stress (*MAPK3K19*) [80]. The loss of melatonin synthesis genes (*AANAT*, *MTNR1A/B*, and *ASMT*) in the sperm whale, killer whale, bottlenose dolphin, and minke whale is associated with unihemispheric sleep [91], which enables one hemisphere of the brain to sleep while the other one coordinates the generation of heat and regular surfacing for breathing [100].

Cetaceans were faced with osmoregulation challenges during their transition to the hyperosmotic marine environment. It has been shown that genes involved in osmoregulation, such as *ACE*, *AGT*, *AQP2*, and *SLC14A2*, are under positive selection in cetaceans, suggesting that they have evolved to maintain the water and salt balance in response to the hyperosmotic environment [101,102]. The osmoregulation-related gene (*UT-A2*) is also under adaptive evolution in cetaceans [103]. Generally, whole-genome sequences of cetaceans exhibit unique features related to various physiological and morphological deviations required for successful aquatic life, characterized by adaptations to physiological stresses due to lack of oxygen, elevated ROS levels, and salt [67]. Cognizant of the important phenotypic modifications in cetaceans, it is thought that the aforementioned modifications have been shaped by natural selection as an adaptation to marine life. Some of the candidate genes for selection in ungulates adapted to marine habitats due to the selective pressures that exist in these environments are summarized in Table 3.

### 3.5. Arctic Habitats

The Arctic is a polar region that is marked by constant light in the summer and nearly complete darkness, low temperatures, and food scarcity in the winter [10]. The woolly mammoth, for instance, lives in the extremely cold, dry steppe-tundra, where the average temperatures are between −30 and −50 °C in winter [11,13]. The ungulates that live in these cold habitats possess different adaptations to the unique challenges and pressures.

Ungulates inhabiting arctic regions have thick and hairy skin due to epidermis and hair structure changes to cope with the extremely low temperatures [104]. Genome sequence analysis of the Yakutan horse, which inhabits the coldest regions in Yakutia, has shown that *BARX2* (involved in hair and epidermis development) and *PHIP* (a key regulator of insulin metabolism) are under positive selection, suggesting an adaptation of ungulates to cold habitats [11]. Similarly, positive selection of temperature-sensitive transient receptor potential (thermoTRP) channel genes (*TRPM8*, *TRPV3*, *TRPV3*, and *TRPA1*) involved in thermal sensation and hair growth in woolly mammoth genome suggests an adaptation to extremely cold conditions in artic regions [13]. Positive selection of fat metabolism genes (*APOB* and *FASN*) in reindeer further supports the assertion that ungulates in artic regions have evolved to survive under cold conditions [12]. Copy number variation of fatty acid metabolism (*ALDH2*) and temperature regulation genes (*ACADSB*, *CYP11B2*, *HSPG2*, and *ATP1A2*) in the Yakutian horse suggests that ungulates have evolved to survive in such harsh environments [11]. Additionally, the *THRAP3* gene, which plays a crucial role in regulating vasoconstriction and vasodilation reflexes in cold temperatures, was reported to be copy number variable in the Yakutan horse [11]. Comparative genome analysis of the reindeer from northern Eurasia and some closely related species has recently shown strong signals of selection in cold-responsive genes (*RPL7* and *SCN11A*) in reindeer [58].

Arctic ungulates also possess adaptations related to their circadian clocks to maintain normal rhythms, notwithstanding the extreme seasonal fluctuations of light and dark in their habitats [11,13,58]. For instance, the comparative genome analysis of the reindeer from northern Eurasia and other ruminant species recently detected several positively selected genes (e.g., *PER2*, *GRIA1*, *NOCT*, *GRIN2B*, *ITPR3*, *GRIN2C*, *NOS1AP*, and *ADCY5*) in the reindeer associated with the circadian rhythm pathway [12]. Similarly, amino acid substitutions specific to the woolly mammoth have previously been shown to occur in genes that maintain normal circadian rhythms, such as *HRH3*, *PER2*, and *HRH1*, suggesting the adaptation of their circadian system to the light–dark extremities/oscillations of the arctic [13]. The positive selection of genes that regulate the circadian clock/rhythm, such as *FXBX21*, *GRIA1*, and *CRY1*, have also been reported in the Yakutian horse [11] and reindeer from northern Eurasia [58], and reindeer from northern China [12].

Although reindeers inhabit arctic regions with extended periods of low or no solar energy, they develop large antlers that require efficient calcium metabolism and reabsorption [12]. Unlike the genomes of other ruminant species, vitamin D metabolism genes (*CYP27B1*, *POR*, *TRPV5*, and *TRPV6*) have been reported to be under selection in the reindeer genome [12,58]. Some of the candidate genes associated with various adaptive responses/features of arctic/subarctic habitats are summarized in Table 3.

### 3.6. Processes Underlying Adaptive Changes in Domesticated Ungulates

Domestication is an evolutionary process by which wild animals are artificially selected to adapt to agricultural environments. The evolutionary basis of domestication is an ancient question, and its genetic basis is becoming more traceable due to the availability of several ungulate genome sequences [105]. A combination of natural and artificial pressures shapes the genetic composition of domesticated ungulates by leaving footprints in the genome that might be detectable [106]. Therefore, it is important to document genomic signatures in domesticated ungulates arising from natural and artificial selection. Genome scans of selection signals have been conducted in several domesticated ungulates, such as the sheep [55,76,105,107,108,109,110,111], goat [4,48,69,105,112,113,114], cattle [41,42,115,116,117,118], pig [77,119,120,121,122,123,124], donkey [56,125], camel [36,38], and horse [44,126,127].

Domesticated environments are more disease-prone and pathogen-infested than wild habitats [49]. Copy number variations are an important basis of immunity development in the domesticated environment [128]. Several investigations have shown an expansion/contraction of immune-related genes in domesticated ungulates, suggesting that selection is probably driven by the various types of pathogens in the domesticated environment, which facilitate natural and artificial selection for adaptive immunity [4,129,130]. For instance, the genome analysis of wild and domestic goats established the acquisition of more copies of immune-related genes (C*FH* and *TRIM5*) in the domestic goat [4]. Copy number variations of immune genes including *ULBP17* in the domestic yak [130], *IL1B*, *CD68*, *CD36*, *CD163*, *IFIT1*, and *CRP* genes in the domestic pig [121], and *CATHL4* and *ULBP1* in cattle [116] have been reported and are suggested to be an adaptive response to various parasites/pathogens. Similarly, duplication of the *KRTAP9-1* gene involved in tick resistance has also been reported in cattle, suggesting an adaptation to parasitic infections in their domesticated environments [98]. Genome comparison of domesticated and wild *Capra* species recently identified the enrichment of several genes, including *SERPINB3*, *SERPINB4*, *CD1B*, *COL4A4*, *BPI*, *MAN2A1*, and *CD2AP*, which are putatively associated with parasite invasion and immunosuppression, an indication that domestication may have impacted pathogen resistance and goat adaptation to the anthropogenic habitats [49]. Additionally, disease-resistant genes such as *CHIA*, *CHI3L2*, *PRDM2*, and *KDM5B* involved in the defense response against nematodes or bacterial infections have been reported to be under selection in sheep [108] and cattle [106]. Furthermore, several defense response genes (*AZU1*, *ELANE*, *GZMM*, *PRSS57*, *PRTN3*, and *CFD*) have been reported to be under strong selection signals in cattle [115]. Genomic variation studies have shown that interferon (*IFN*) gene families that are mainly involved in the defense against viral infections and other immune response genes such as *NPG3*, *PMAP23* and *CAMP* are copy number variable in the pig [131]. Fertility and productivity are some of the most important factors in breeding and domestication [120]. The analysis of signatures of selection associated with economically important traits have been done in various populations/species of domesticated ungulates, such as pigs [132], goats [69,114], and sheep [69]. For example, genes associated with productivity and fertility traits, such as the *MTMR7* gene, is under positive selection in sheep and goats [113]. The pig genome has shown signatures of selection for *FASN* and *MOGAT2* genes that code for a fatty acid synthase for increased fat absorption/digestion and total body weight [132]. Signatures of selection at the *PLAG1*, *LCORL*, and *NR6A1* loci have also been identified across commercial lines of European domestic pigs reared for meat [123]. In cattle, several productivity genes, such as *GBP4*, *FGF6*, *TIGAR*, and *CCND2* (involved in body weight and stature), *TMEM68*, *TGS*, *LYN*, *CEP72*, and *SLC9A3* (involved in growth and feed efficiency), and *GID4*, *ATPAF2*, *ELOVL3*, and *NFKB2* (involved in milk yield and components), have also been reported to be under accelerating evolution [106]. Strong signals of selections in the *ABCG2* gene, which is associated with milk yield composition, have also been reported in goats [112], cattle [133], and sheep [134]. Genes involved in wool production (*LCE7A* and *MOGAT*) have been reported to be under positive selection in domesticated sheep [53]. The positive selection of the *KITLG* gene which determines litter sizes has also been identified in sheep (*O. aries*) and goats (*C. hircus*), and could also have facilitated their initial domestication due to productivity [105]. In the yak genome, reproduction-related genes (*AKR1C3*, *IZUMO1*, and *TSEG2*) and growth-related genes (*KLF6*, *GPC1*, *CHRM3*, and *CHKB*) were shown to be under copy number variations [130].

Coat coloration is one of the important phenotypic traits of domestic ungulates over their wild relatives/ancestors [135]. According to Fontanesi et al. [136], coat coloration is determined by various genes that influence the occurrence, biochemical activities, and distribution of melanocytes. While wild ungulates tend to have uniform colors and patterns per species, their domesticated counterparts display various colors and patterns [4]. Evidence suggests that coat colors are largely influenced by duplication of the *ASIP* gene in sheep [137,138] and goats [136]. Other coat color genes, such as *GNAQ*, *ATRN*, *HELLS*, *OSTM1*, *MUTED*, *TRPM7*, *ADAMTS20*, *VPS33A*, *MITF*, *SLC7A11*, and *OCA2*, have also been shown to be copy number variable in the domestic goat as opposed to their wild relatives, suggesting an adaptive phenotype which arose during domestication [4]. In sheep, the duplication of *ASIP* has previously been associated with the typical white coat colors [137], which are generally favored in sheep for wool production [128]. Several other studies have illustrated the positive selection of other genes related to coat color in domestic ungulates, for instance, *KIT* and *DUP4* in pigs [123], *KIT*, *MC1R*, *BRM*, *ASIP*, and *FGF5* in cattle [139], *MITF* and *EDNRB* in Chinese domestic pigs [140], *IRF4*, *EXOC2*, *RALY*, *EIF2S2*, and *KITLG* in domestic goat breeds [48], and *MC1R* in certain horse breeds [126].

Tameness and less aggressiveness are also associated with domesticated ungulates over their wild counterparts, partly reflecting their adaptations from historical associations with humans [141]. Several PSGs, such as *HTR3A* related to the release and concentration of serotonin in the central nervous system, which is strongly linked to behavior, were previously identified in the domestic goat, suggesting their roles in the selection of less-aggressive behaviors and tameness during its domestication [4]. Signals of selection in behavioral genes (*ATP2B2*, *SCRIB*, *ASPA*, *MYO6*, *NTRK2*, *PLXNB1*, and *SNCA*) have also been reported in the domestic yak [142]. High copy number differentiation of Neural development genes (*GRIN2D*, *KCNJ14*, *SHANK3*, *NTF4*, and *CA11*) between the domestic and wild yak is suggested to underlie the processes that contributed to the successful domestication of the yak [130].

Artificial selection has also left important selective footprints throughout the genomes of domesticated ungulates. For example, a recent genomic study of the domestic goat showed that neural processes genes (*RRM1* and *STIM1*) that control behavior were subjected to artificial selection during domestication [49]. Similarly, genome sequence analysis has shown that the racing gene (*MSTN*) and other performance-related genes such as *ECA23* were subjected to artificial selection in domesticated horses, thus explaining their enhanced performance in racing [126,127]. Other studies have also demonstrated the positive selection of genes that regulate various reproductive traits (*PRM1*, *PRM2*, *TNP2*, *GPR149*, and *JMJD1C*) in pigs [140]. Numerous positive selected genes related to production traits, such as meat yield (*CHRM3*, *TMEM186*, *PISD*, *DES*, and *PPARGC1A*) [143,144,145], dairy traits (*ITFG1* and *SLC27A6*) [139,146], and reproduction (*NKD1*, *SPAG4*, and *ATP2B1*) [147,148,149], which have been artificially selected were reported in several studies in cattle and other ungulate species.

Adaptive introgression also contributes to beneficial phenotypes seen in domesticated ungulates. For example, adaptive introgression of the *MUC6* gene from wild goats into domesticated goats is associated with gastrointestinal pathogen resistance [49]. Similarly, introgression of the *PAD12* gene from wild sheep species to the domesticated sheep is associated with enhanced resistance to pneumonia [150]. Strong signals of adaptive introgression of the oxygen transport system and sensory perception genes (*HBB* and *RXFP2*) have also been reported in Tibetan sheep [151]. Adaptive introgression have also been reported in other domesticated ungulates, such as cattle [152,153], pigs [124], and camels [154]. Adaptive responses to domestication as selective pressure and the associated candidate genes that have been associated with domesticated ungulates by several researchers are captured in Table 3.

### 3.7. Processes Underlying Adaptive Changes in Feral Ungulates

Feral ungulates are animal species that escaped from captive environments into the wild. Genomes for feral ungulates are also shaped by natural and artificial selection and adaptive introgression. Adaptive introgression of major histocompatibility complex (MHC) DRB in the Alpine ibex from the domesticated goat is suggested to confer an improved immune response by broadening the MHC sequence repertoire [155]. The *RXFP2* gene that controls horn size is under positive selection in semi-feral sheep and is associated with the acquisition of weapon-grade horns due to sexual selection and minimal human intervention [109]. Feral Andean horses have been shown to have undergone extensive selection pressures to adapt to high altitudes [156]. Although feralization is associated with adaptations in ungulates, it also contributes to maladaptation [141]. Feral ungulates provide unique opportunities to understand the genomics impacts of domestication. Hence, there is a need to conduct more genome scan screening of selection signatures in feral ungulates to utilize their rich genetic resources.

**Table 3 animals-11-01617-t003:** Candidate genes identified in ungulates, proposed selection pressures, and adaptive responses.

Environment	Selective Pressure (s)	Selection Signature	Genes under Selection in Ungulates	Biological Functions of Candidate Genes
Desert/Arid	Low water availability	Population differentiation and increased frequency of derived alleles	Red deer (*CP2U1*) [3], Camel (*CYP2J*, *CYP2E*, *AQP1*, *AQP2*, *AQP3*) [5,36], sheep (*NXA6*, *GPX3*, *GPX7*, *PTGS2*, *CPA3*, *CPVL*, *ECE1*, *CALM2*, *CACNA2D1*, *KCNJ5*, and *COX2*) [55]	Water-salt balance, regulating water retention and reabsorption
	Airborne dust and allergic diseases	Population differentiation, excess of long haplotype and Sequence altering mutant (rapidly evolving genes)	Red deer (*TRAF2* and *IL1R1*) [3], camel (*FOXP3*, *CX3CR1*, *CYSLTR2*, and *SEMA4A*) [5], Fat-tail sheep (*ZBP1*, *PRDX1*, *MAST2*, and *LURAP*) [71], Bakri goat and sheep (*GRIA1*, *IL2*, *IL7*, *IL21*, *IL1R1*) [69]	Defend against airborne dusts
	High UV exposure	Excess of long haplotypes and rapidly evolving genes	Red deer (*LAMB1*, *LAMB2*, *CYC*, *FANCF*, and *GPR98*) [3], camel (*OPN1SW*, *CX3CR1*, and *CNTFR*) [5]	Ocular development, visual protection, and photoreceptor cell synapses
		Excess of long haplotypes	Red deer (*SLX4*, *FANCF*, *FANCG*, *FANCI*, *ATR*, and *POLH*) [3], Fat-tail sheep (*PMS1*, *SPO11*, *RAD54L*, *MUTYH*, *CHEK2*, *POLR2D*, and *CMPK1*) [71]	DNA repair
		Sequence altering mutant (rapidly evolving genes)	Desert goat (*ABCA12*, *ASCL4*, and *UVSSA*) [72]	Skin barrier development and function
	High temperature	Population differentiation and excess of long haplotypes	Red deer (*GNAI2*, *FZD4*, *MP2K2*, *CREB3*, *CBP*, *GNAO*, *TF7L2*, and *GNAO*) [3], goat (*MTOR*, and *MAPK3*) [114], Fat-tail sheep (*ERCC3*, and *TGM3*) [71], Bakri goat and sheep (*FGF2*, *GNAI3*, and *PLCB1*) [69]	Response to thermal stress
	Xenobiotic compounds	Population differentiation and excess of long haplotypes	Red deer (*CP2U1*, *CP3AS*, and *CP3AO*) [3]	Plant secondary metabolism
Arctic	Long light and dark periods	Population differentiation, rapidly evolving genes and copy number variable genes	Yakutian Horse (*LECT2*, and *FBXL21*) [11], Wooly mammoth (*HRH3*, *PER2*, and *BMAL1*) [13], reindeer (*GRIA1*, and *OPN4B*) [12,58]	Regulation of the circadian clock
	Low temperatures	Population differentiation and copy number variable genes, Sequence altering mutant (rapidly evolving genes)	Yakutian Horse (*ACADSB*, *ATP1A2*, *CYP11B2*, *HSPG2*, and *PRKG1*) [11], Wooly mammoth (*DLK1*, and *TRPV3*) [13], Yakutian cattle (*DNAJC9*, *SOCS3*, *TRPC7*, *SLC8A1 GLP1R*, *PKLR*, and *TCF7L2*) [42], Reindeer (*SCN11A*, and *SILT2*) [58]	Thermoregulation
	Lipid metabolism	Sequence altering mutant (rapidly evolving genes)	Reindeer (*APOB* and FASN), woolly mammoth (*CRH*) [13]	Lipid metabolism
High-altitude	Hypoxia	Population differentiation, sequence altering mutant (rapidly evolving genes), excess of long haplotypes	Tibetan wild boar (*ALB*, *ECE1*, *GNG2*, and *PIK3C2G*) [52], Yak (*ADAM17*, *ARG2*, and *MMP3*) [8], goat (*CDK2*, *SOCS2*, *NOXA1*, and *ENPEP*) [157], Tibetan sheep (*EPAS1*, *CRYAA*, *LONP1*, *NF1*, *DPP4*, *SOD1*, *PPARG*, and *SOCS2*) [78], Tibetan pig (*EPAS1*, *HIF1A*, *RNF4*, *TNFSF1*, *PDE1A*, and *PDE3*) [77]	Hypoxia response
	Low temperature	Population differentiation and sequence altering mutant (rapidly evolving genes)	Tibetan wild boar (*AEBP1*, *DGAT1*, *FABP2*, *LEPR*, and *PTPN1*) [52], Yak (*GCNT3*, *HSD17B12*, *WHSC1*, and *GLUL*) [8], Tibetan sheep (*DPP4*, and *PPARG*) [78]	Tolerance to cold
	Intense UV radiations	Sequence altering mutant (rapidly evolving genes)	Tibetan wild boar (*BCL3*, *ERCC4*, *ERCC6*, *REV1*, *USF1*, and *ZRANB3*) [52]	DNA repair, response to radiations
Marine	Low oxygen levels	Sequence altering mutant (rapidly evolving genes)	Whales (*PRDX1*, *PRDX2*, and *GPX2*) [96], Minke whale (*GPX2*, *ODC1*, *GSR*, *GGT6*, *GGT7*, *GCLC*, and *ANPEP*) [67], cetaceans (*ALDOA*, *ENO2*, *CS*, *ATP6V0A4*, *LHPP*, *NDUFA9*, and *NDUFV3*) [158]	Response to hypoxia
	Salty water	Sequence altering mutant (rapidly evolving genes)	Minke whale (*AGTR1*, *ANPEP*, *LNPEP*, *MME*, and *THOP1*) [67], dolphin (*TSPO2*, *EPGN*, *PLN*, *EDN2*, *PLA2G5*, and *KCNJ2*) [15]	Salt water balance
	Aquatic environment	Sequence altering mutant (rapidly evolving genes)	Minke whale (*HOXA5*, *HOXB1*, *HOXB2*, *HOXB5*, *HOXD12* and *HOXD13*) [67]	Morphological adaptation to swimming
	Prolonged, deep diving	Sequence altering mutant (rapidly evolving genes)	Whales and dolphins (*GSTA1*) [96], dolphin (*APOA2*, *APOC4*, *APOO*, *FABP4*, *SERINC4*, *CCDC129*, *PLA2G5*, *PNLIPRP3*, *RARRES2*, and *NR1I3*) [15], cetaceans/whales (*LDHA*, *LDHD*, *PC*, *PCK1*, *FBP1*, and *GPI*) [158]	Energy metabolism
	Cold temperature	Sequence altering mutation (rapidly evolving genes)	Minke whale (*NPY*) [66]	Thermoregulation
Domesticated	Productivity	Population differentiation, rapidly evolving genes, copy number variable genes	Goat (*LRP1*, *PLIN4* and *FASN*) [4], pig (*ACACA*, *ANKRD23*, *GM2A*, *KIT*, *MOGAT2*, *MTTP*, *FASN*, *SGMS1*, *SLC27A6*, and *RETSAT*) [132], donkey (*TBX3*, *NCAPG*, *LOCR*, *BCOR*, *CDKL5*, and *ACSL4*) [125], cattle (*MUC*1) [42]	Regulation body weight, body size, milk production
	Domestication	Population differentiation, sequence altering mutant (rapidly evolving genes), copy number variable genes and excess of long haplotypes	Goat (*FGF9*, *IGF1*, *ASIP*, *KITLG*, *HTT*, *GNA11*, *OSTM1*, *ATRN*, *GNAQ*, *HELLS*, *MUTED*, *VPS33A*, *ADAMTS20*, *MITF* and *OCA2*) [4,114,145], donkey (*ASIP* and *KTLG*) [125], sheep and goat (*KITLG*, *HMGI-C*, *NBEA*, and *MTMR7*) [105], pigs (*ESR1*, and *AHR*) [159,160]	Coat color, litter size, fatty acid composition, wool crimping
	Domestication	Population differentiation, excess of long haplotypes, sequence altering mutant (rapidly evolving genes), and copy number variable genes	Goat (*HTR3A*, *STMI*, and *PRMI*) [4,49], *horse* (*VDAC1*, and *GRID1*) [127], pigs (*NRTN*, *SEMA3C*, *PLXNC1*, *AAK1*, *RAB35 FRS2*, *APBA2*, *MC4R*, *RCAN1*, and *BAIAP3*) [161] and domestic yak (*GRIN2D* and *NTN5*) [130]	Tameness, less aggressiveness, reduced fear to humans
	Increased pathogens	Excess of long haplotypes, sequence altering mutant (rapidly evolving genes), population differentiation and copy number variable genes, adaptive introgression	Goat (*IL10RB*, *IFNLR1*, *BCL2L1*, *ERBB2*, *ENO1*, *CFH*, *TRIM5*, and *MUC6*) [4,49,114], cattle (*IFNAR1*, *IFNAR2*, *IL10RB*, *NOD2*, *CD96*, *CD14*, *GZMB*, *IL17A*, *PFKM*, *ADAM17*, *SIRPA IFNAR2*, *IFNG*, *CD34*, *TREM1*, *TREML1*, *FCER1A*, *IL23R*, *IL24*, *IL15*, and *LEAP2*) [41,42], pig (*IL1B*, *CD36*, *CD68*, *CD163*, *CRP*, and *IFIT1*) [121]	Host innate immune response, gastrointestinal pathogen resistance, disease resistance

## 4. Future Perspectives and Directions

Adaptive evolution studies in ungulate genomes have provided valuable insights into their genetic background and the adaptation to their respective environment. Nevertheless, there is room for deepening insights into these important and complex genomic bases of evolutionary adaptations, given the opportunities presented by next-generation sequencing advances. There are limited studies on adaptive evolution between species compared to within species; therefore, it is necessary to conduct more interspecies studies to understand the adaptation of diverse ungulates to a given environment. For instance, it will be interesting to compare the genomes of desert ungulates versus those inhabiting different environments to determine convergent evolution. The majority of reported adaptive evolution studies focused on recent selection signatures within populations; hence, there is a need to identify selection signatures that occurred before species divergence from their ancestors. Having identified loci thought to be involved in adaptation, it would be desirable to further confirm the fitness effects of selected genes in ungulates using experimental methods, such as gene expression analysis. Advances in genome editing technologies hold great promise as ungulates genetic improvement tools. Genome editing technologies such as clustered regularly interspaced short palindromic repeats (CRISPR/Cas9) could be used to make specific and precise changes to ungulate genomes (especially those of the domesticated species) to improve their resilience to stressors in their environments or to enhance their adaptation to changing climates. Climate change is a reality affecting both the environment and the animals that live in it. Genomic approaches have been one of the tools used to either mitigate or adapt to climate change. Adaptive evolutional studies in ungulates, particularly from different environment conditions presents an opportunity to better understand genes and loci under selection in response to climatic change and other environmental factors that can be used in efforts to mitigate the effects of climate change or adapt animals to effects of climate change.

## 5. Conclusions

Ungulates are a diverse group of animals that exist in some of the most extreme habitats on earth. Although both artificial and natural selections have left discrete signatures on their genomes and several candidate genes identified, an account of the fitness effect of the candidate genes involved in adaptive responses to the selective pressures in different ungulate habitats is largely lacking. The present review has assessed the evolutionary genetic signatures underlying the morphological and ecological diversity of ungulates in different habitats. It has highlighted the tremendous efforts made in understanding the genomics of adaptation in ungulates and the opportunities which remain unexplored.

## Figures and Tables

**Figure 1 animals-11-01617-f001:**
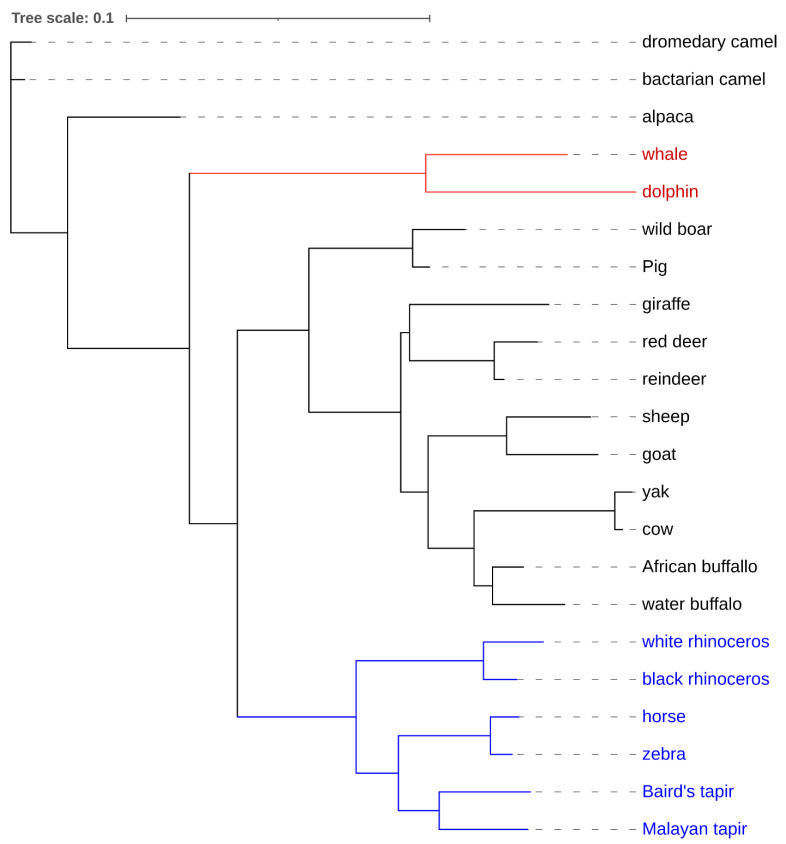
Cladogram for ungulates. Phylogenetic tree for selected even-toed ungulates, including cetaceans and odd-toed ungulates. Odd-toed ungulates are labeled in blue, while cetaceans are in red. The phylogenetic tree was inferred from sequence data for selected ungulates Cytochrome b (Cytb) gene downloaded from genbank [17]. Multiple sequence alignment was carried out using MUSCLE [18], while the phylogeny tree was inferred using PhyML 3.0 [19] and visualized in the Interactive Tree Of Life tool [20].

**Table 1 animals-11-01617-t001:** Population genomic approaches for identifying selection signatures.

Type of Signature	Detectable Pattern	Methodologies
Change allele frequency spectrum	Increased frequency of derived alleles	Tajima’s D [24]
Extended haplotype homozygosity	Linkage disequilibrium (LD) persistency and unusual long-range haplotypes	Cross-population extended haplotype homozygosity (XP-EHH) [23]
		Integrated haplotype score (iHS) [23]
Population differentiation	Different allele frequencies between populations	F_ST_ [23]
Composite methods	Detects increased frequency of derived alleles, difference in allele frequencies and unusual long-range haplotypes	CMS [26]

**Table 2 animals-11-01617-t002:** Publicly available genome sequences of ungulates.

Ungulate	Species	Genome Size (Gbp)	Number of Annotated Genes	GenBank Assembly Accession
Desert ungulates	Bactrian camel	2.4	20,251	AGVR01000000, JARL00000000
	Dromedary camel	2.5	20,714	JDVD00000000
	Red deer	3.3	22,138	MKHE00000000
High-altitude ungulates	Wild yak	2.6	22,282	AGSK01000000
	Siberian musk deer	3.1	19,363	GCA_011751665.1
Artic ungulates	Rein deer	2.9	27,332	GCA_014898785.1
Savanna	Giraffe	2.9	17,210	LVKQ00000000
	African buffalo	2.7	19,765	SAMN05717674
Marine	Blue whale	2.4	19,518	GCA_009873245.2
	Dolphin	2.5	16,550	GCA_011057625.1
Domesticated ungulates	Cow	2.7	21,880	GCA_002263795.2
	Sheep	2.9	20,506	GCA_002742125.1
	Goat	2.9	21,361	GCA_001704415.1
	Pig	2.5	21,303	GCA_000003025.6
	Donkey	2.3	19,963	GCA_003033725.1
	Horse	2.5	20,955	GCA_002863925.1

## Data Availability

Not applicable.

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
