# Peer review of "Genomics of Adaptations in Ungulates"

_animals, 2021, doi:10.3390/ani11061617_

Round 1
Reviewer 1 Report
The authors wrote a comprehensive review on genomics of adaptations in ungulates from different habitat such as desert and arid, arctic, high altitude, marine and domestic setting. The topic is very interesting and beneficial for a broad range of readers who are interested in the effect of natural selection on candidate genes underlying adaptive responses to the selective pressures in different habitats.
There are two main suggestions, and few formatting corrections as below:
- Selective pressures in domestic setting
Considering domestic setting could be in any of the various environment mentioned above, I would discuss it as a process underlying adaptive changes, rather than a “habitat”. Considering domestication is the result of a complex interplay of biological, environmental and cultural processes, natural and artificial selection both play an important role in selection of candidate genes. Therefore, I would recommend to add few lines discussing natural selection underlying genetic adaptation to various environmental conditions as well as artificial selection on phenotypic traits related to domestication syndrome, and economically valuable traits. I appreciate that the review paper is about genomic signature of natural selection in response to environmental conditions, but if domestication is mentioned one should not ignore the important of artificial selection. For example, selection on MSTN gene for racing performance in horses is the result of extensive breeding program over the last 200 years, and not necessarily adaptation to the habitat. Therefore, listing them all in table 3 as adaptive responses in different habitat is not accurate.
- Feral ungulates
In addition, a section should be allocated to genomic signature underlying adaptation or maladaptation in feral animals. This is important as many of the ungulated are indeed feral. Moreover, feral animals often introduce artificially selected traits into their invaded environment, which might modify natural ecosystems, or force adaptation/maladaptation in feral species itself.
- Few minor comments
Please list the references numerically throughout the manuscript.
Line 47:
For better resolution, perhaps download the data and reconstruct the tree. In addition, it’s important to mention the phylogenetic trees are constructed based on genetic markers. Although not necessary, adding pictures of the ungulates on the tip of the branches of the phylogenetic tress would make it much nicer to look at.
Line 47:
Replace “Phylogeny tree” with Phylogenetic tree
Replace “even-toe ungulates” with even-toed ungulates
Line 48:
Replace “Phylogeny tree” with Phylogenetic tree
Replace “odd-toe ungulates” with odd-toed ungulates
Line 52:
Replace “to comprehending” with to comprehend
Line 53-54:
Delete the sentences below, as it is repetitive of the previous lines.
“Cognizant of this, genomics can be used to understand the adaptive evolution of ungulates in different habitats”.
Line 75:
Replace “score (iHS) [21].” With score (iHS) [21]
Replace “FST [19].“ with FST [19]
Line 105:
Add below references for camel genomes as they are important studies.
- The first African dromedary camel genome reference sequence:
Fitak RR, Mohandesan E, Corander J, Yadamsuren A, Chuluunbat B, Abdelhadi O, Raziq A, Nagy P, Walzer C, Faye B, Burger PA. Genomic signatures of domestication in Old World camels. Commun Biol. 2020 Jun 19;3(1):316. doi: 10.1038/s42003-020-1039-5. PMID: 32561887; PMCID: PMC7305198.
- Genomic signature of selection in old world camels:
Fitak, R. R., Mohandesan, E., Corander, J. & Burger, P. A. The de novo genome assembly and annotation of a female domestic dromedary of North African origin. Mol. Ecol. Res 16, 314–324 (2016).
- The dromedary camel genome annotation by Chromosome:
Elbers JP, Rogers MF, Perelman PL, Proskuryakova AA, Serdyukova NA, Johnson WE, Horin P, Corander J, Murphy D, Burger PA. Improving Illumina assemblies with Hi-C and long reads: An example with the North African dromedary. Mol Ecol Resour. 2019 Jul;19(4):1015-1026. doi: 10.1111/1755-0998.13020. Epub 2019 May 17. PMID: 30972949; PMCID: PMC6618069.
Line 118:
Table 2: The GenBank assembly accession”JDVD00000000” reported for dromedary camel but the reference is missing. Add the reference Elbers et al. (2019)
Line 138-139:
Remove the phrase below as it is repetition – add the reference 63 to the line 133.
Long-term exposure to intense UV radiations is known to contribute to ophthalmic 138 conditions [5,63].
Line 271:
Replace “several decades ago” with some million years ago.
Line 284: remove the space before SCN4A
Line 285: replace “the cow” with cow
Line 304: remove the space after ROS
Line 404: remove the space before disease-resistant
Line 419-421:
Move the paragraph below after the line 413 as that is relevant to the reproduction traits and fertility.
Other genes under positive selection for productivity and fertility traits in domesticated ungulates include the MTMR7 gene in sheep (O. aries) and goats (C. hircus) that regulate fatty acid composition.
Line 442:
Replace “have been reported to be” with are
Reviewer 2 Report
This paper represents a successful attempt at combining most of the available findings in terms of adaptive genes across several species. The paper is very well organised and concepts are conveyed clearly. The English form is particularly good. I just have a couple of suggestions to broad the reach of this paper. I would encourage the authors to consider the analysis of adaptive introgression by means of local ancestry inference as a strategy to detect putative adaptive genes in several domestic species as sheep cattle. I believe this would make an interesting paragraph that would attract more readers. Finally, I would point out that Figure 1 seems to have very low quality. If this is not a ‘review version’ dependent issue, than it is necessary to generate a higher quality image.
